# Micromagnetics of Microwave-Assisted Switching in Co-Pt-Based Nanostructures: Switching Time Minimization

**Christos Thanos [1] and Ioannis Panagiotopoulos [1,2,*]**

1    Department of Materials Science and Engineering, University of Ioannina, 45110 Ioannina, Greece
2    Institute of Materials Science and Computing, University Research Center of Ioannina (URCI), 45110 Ioannina, Greece
\*    Correspondence: ipanagio@uoi.gr; Tel.: +30-2651007182

**Abstract:** Microwave-assisted switching (MAS) is simulated for different CoPt and CoPt/Co$_3$Pt nanosrtuctures as a function of applied DC field and microwave frequency. In all the cases, the existence of microwave excitation can lower the switching field by more than 50%. However, this coercivity reduction comes at a cost in the required switching time. The optimal frequencies follow the trends of the ferromagnetic resonances predicted by the Kittel relations. This implies that: (a) when the DC field is applied along the easy axis, the coercivity reduction is proportional to the microwave frequency, whereas (b) when the coercivity is lowered by applying the DC field at an angle of 45° to the easy axis, extra MAS reduction requires the use of high frequencies.

**Keywords:** microwave-assisted switching; microwave-assisted magnetic recording; magnetization reversal; micromagnetic simulation

## 1. Introduction

Microwave-assisted magnetic recording (MAMR) refers to the application of microwave-assisted switching of the magnetization (MAS) in recording media. In MAS, the switching field is lowered by resonantly exciting the precessional motion of the magnetic moment by a radio frequency (rf) field. The MAS response is fundamentally different from the thermal response. MAMR seems to be the most promising tera-bit-class magnetic recording technology today [1]. Using MAS, the required writing field at the head can be reduced, thus alleviating the conflicting requirements of increased density, high SNR, writability, and thermal stability [2–5]. The efficiency of MAS on nanoparticles has been demonstrated long ago [2,6]. As it regards perpendicular recording media, it has been experimentally shown that using MAS, the coercivity can be measurably reduced over a wide range of frequencies, and that it has a definite minimum [7]. In CoCrPt-SiO$_2$ granular medium [8], it was observed that the coercivity decreases linearly with an increase in microwave frequency, and the coercivity reduction ratio can be as high as 80%.

The feasibility of MAMR is based on the development of Spin-Torque Oscillators [9–13]. Compared to other means of dealing with the writability problem, MAMR requires minimal change on the existing digital magnetic recording technology. Furthermore, it can be combined with other ideas that have been proposed to push up the physical limits of areal density as composite hard/soft media [14–16], antiferromagnetically coupled media [17,18], and tilted media [19–23]. The selectivity of this resonance-based process, makes it applicable to 3D data storage architectures [1,16,24] already introduced to competitive NAND flash technology.

One of the most fundamental conflicts in recording media requirements is that the thermal stability imposes the use of high-anisotropy materials, creating writability problems. It is less stressed though that on the other hand, in high anisotropy materials, the frequencies of the precessional motion are higher, making the switching processes faster. Reversal

time (switching time) is an important parameter that has been less studied in the literature [16,25,26]. The switching time in composite media drops from several nanoseconds to a fraction of a nanosecond as the field is increased to a value double of the coercivity, similar to what is observed in single-phase media [14]. Switching at lower applied fields with high frequency MAS seems to come at a price of longer switching times [2]. In Fe-Co thin films, the switching field is saturated for microwave durations above 50 ns, an effect attributed to the competition between the pumping and damping processes [27]. It has been reported that in micron-sized elements, where the reversal is inhomogeneous (i.e., involves nucleation–propagation–relaxation processes, mechanisms with different relevant frequencies), the switching time is a complex oscillating function of the microwave frequency [28]. It must be noted, however, that the dynamics that govern the MAS reversal are complex anyway, even within a one-macrospin approximation: the process depends sensitively to all the parameters involved [29], including the rise time and duration of microwave pulses [26,30]. This can be simply understood by considering that during the switching process, the effective field which depends on the magnetic state, changes during the magnetization reversal, and so does the precession frequency. Therefore, the optimal effect may be achieved using "chirped" microwaves [31–33], in which the frequency is varied to match the varying magnetization precession frequency, a technologically challenging task.

In the following, we present micromagnetic simulations of the microwave-assisted reversal of disk-shaped nanoelements and compare the effect of size and angle of the DC field. For the main part, we have considered fixed frequency rf fields of 1 ns duration, and we discuss the issue of pulse duration and rise time in the last section. As typical examples, we have considered single-layer CoPt and CoPt/Co$_3$Pt nanodisks. Co-Pt alloys are typical materials proposed for high-density magnetic recording media [34–36]. In particular, the equiatomic chemically ordered CoPt (as well as FePt) alloys consist of alternative Co and Pt layers along the c-axis of the tetragonal L1$_0$ structure. This atomic arrangement gives high anisotropy [37]. These high anisotropy materials can be combined with semihard phases such as the, also chemically ordered, Co$_3$Pt to reduce the coercive field in favor of writability [38]. The key advantage of such composite structures is that the reversal process which is induced by thermal activation, is more homogeneous than the reversal induced by applying an external field. Thus, due to the different reversal modes of the field-induced switching process and the temperature-induced switching process, the ratio of the energy barrier over the coercive field can be optimized to achieve high thermal stability without the loss of writability [14]. An extra interesting fact about these phases is that they can be produced with intermediate degrees of chemical ordering, achieving tailor-made properties [39]. The micromagnetic simulations have been performed using the mumax3 package [40,41]. This is an open-source graphics-processing-unit accelerated micromagnetic simulation package that gives possibility to perform faster, larger, and more complex simulations. The results of the quasistatic properties can be compared with the predictions of the Stoner–Wohlfarth model: this is an exactly solvable model for coercivity based on the simplifying assumption of homogeneous reversal in single-domain particles with uniaxial anisotropy [42]. Assuming homogeneous magnetization, the multiparameter problem of a spatially inhomogeneous magnetic state is reduced to finding the minimum of one-parameter free energy function. This parameter is the angle of the magnetization with the easy axis, which also defines the angle to the applied field. The Stoner–Wohlfarth model has large applicability despite the fact that, even in single-domain particles, the reversal might not be uniform. However, it turns out that non-homogeneous modes have nucleation fields with a similar angular dependence [43]. In order to study cases with clear deviations from homogeneous reversal, we have also modeled 60 nm diameter disks. As MAS is based on the resonant excitation of the precessional motion, we compare the results with the predicted homogeneous resonances of the system given by the Kittel equations [42]. For a uniaxial particle with effective anisotropy $H_K$ along its symmetry axis, under an inversed field $H$ applied along its axis, the Kittel equation is simply $f = \gamma(H_K - H)$, where $\gamma$ is the gyromagnetic ratio, typically $\gamma$ = 28.025 GHz/T. Note, that in this case, the anisotropy

field also defines the Stoner–Wohlfarth coercivity. The corresponding equation for the field applied at an angle of 45° to the easy axis is derived in Appendix A.

## 2. Micromagnetic Simulation Details

The nano-element geometry was that of a disk consisting of either a single hard phase (with the CoPt-type phase parameters saturation magnetization $M_S$ = 800 kA/m and uniaxial anisotropy $K_{mc}$ = 4.9 MJ/m³ [42]) or two phases, where a thin semihard layer (Co₃Pt type with saturation magnetization $M_S$ = 1114 kA/m and uniaxial anisotropy $K_{mc}$ = 0.6 MJ/m³ [42]) is on top of the hard layer. The magnetocrystalline easy axis was set either along the disk normal (z-axis) or at 45° to the disk normal for the tilted media. A small misalignment of 1 deg was introduced to avoid numerical errors that would arise in the cases where the axes of the magnetocrystalline, shape anisotropy, and applied field all coincide. A perfect alignment, apart from not being relevant to real conditions, leads to zero torque: in this case, the reversal is mainly governed by some incubation time related to the random thermal motion of the magnetization.

The thickness of the hard phase layer was $t$ = 4 nm. For the two-phase disks, an extra 2 nm of the semihard phase was added on top of the hard. MAMR was studied for two different diameters: $D$ = 16 nm (which shows coherent reversal) and $D$ = 60 nm, for which highly inhomogeneous reversal occurs. The exchange stiffness was set to $A_{ex}$ = 10 pJ/m. The lateral (along the disk diameters) cell size was set to 1.0 nm, whereas along the disk axis, it was set to 0.5 nm. These values are much smaller than the characteristic exchange length scale, $L_{ex} = \sqrt{2A_{ex}/\mu_0 M_S^2}$, which is close to 5 nm for both phases. This choice ensures the minimization of discretization errors. One way to check this issue is by ensuring that further reduction does not change the results (errors are smaller than the used data point symbols). Furthermore, the maximum angle between two simulation cells can be checked at all times. We have found that this value (which is maximized close to the coercive field) at all cases is less than 0.009 rad for single-phase CoPt materials and less than 0.2 rad at the interface of bilayer CoPt/Co₃Pt nanodisks.

For the dynamic properties, the damping constant was set to $\alpha$ = 0.02. The time discretization of 0.1 ps corresponds to 10 THz, which is well above the frequency response of the systems. For each geometry, the sample was initially relaxed in zero field from a state magnetized along the +z direction. Then, a reversed DC field was applied simultaneously with a circularly polarized alternating field with a frequency up to 420 GHz and the magnetization was monitored for 14 ns. The pulse rise time and duration were systematically varied for selected cases. The assumption of having a perfect material with the bulk material constants could be questioned. In general, for nano-materials in the regime of homogeneous reversal, the presence of imperfections facilitates nucleation and tends to decrease the switching field, whereas in the regime of inhomogeneous reversal, imperfections provide hindrances in the domain wall motion and increase the switching field [42]. Here, we have also included systems above the threshold of the homogeneous reversal, but without introducing any type of imperfection. The intention of this study is not to obtain results for a specific recording medium product, but rather the effect of the main design parameters and their tradeoffs in the MAS process; note that successful simulations of the MAS behavior can be obtained even in the one-macrospin limit [29].

## 3. Results

The starting case is a single hard-phase disk with $D$ = 16 nm, $t$ = 4 nm. The data are shown in Figure 1. The x-axis represents the applied reversed DC field and the y-axis the frequency of the RF field.

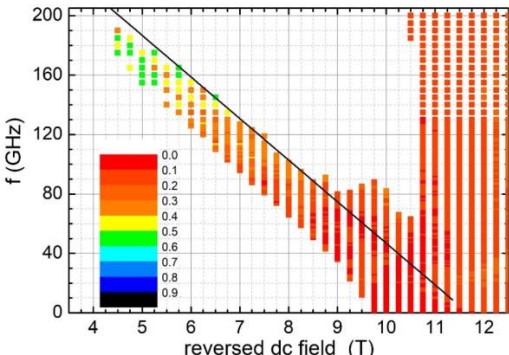

**Figure 1.** Conditions (exciting frequency vs. applied reversed field) under which reversal occurs within 1 ns for a CoPt disk with diameter $D$ = 16 nm and thickness $t$ = 4 nm. The color code represents the switching time in nanoseconds. A particular color is used for switching time, which is between the values indicated at its two edges. The solid black line represents the Kittel condition $f = \gamma\,(11.7\ T - H)$.

The points corresponding to conditions for which magnetization reversal occurs within 1 ns are denoted. These points fall below a straight line corresponding to the ferromagnetic resonance (FMR) frequencies given by the Kittel condition $f = \gamma\,(H_{dc} - H)$, where $H_{dc}$ is the coercivity without MAS. The color code represents the time of the reversal. For this disk geometry, the demagnetization factor along the z direction is $N$ = 0.7341. For a hard phase with anisotropy $K_{mc}$ and saturation magnetization $M_S$, the coherent rotation coercivity is given by the relation:

$$H_{dc} = \frac{2K_{mc}}{\mu_0 M_S} + \frac{(1 - 3N)}{2} M_S, \tag{1}$$

For the used parameters of the CoPt, this is expected to be $H_{dc}$ = 11.7 T, which agrees with the data of Figure 1.

An impressing reduction down to 4.5 T (by 62% of the initial $H_{dc}$) can be achieved if high frequencies (180 GHz) are available, but we must also note that in this case, the reversal time increases to 0.5 ns, compared to just 0.1 ns at higher fields. The reversal can remain fast (0.13 ns) and still have a reduction by 40% of $H_{dc}$ with 100 GHz.

The data are compared with the case where an extra-soft layer is used to reduce the coercivity. By adding a softer $Co_3Pt$ layer of 2 nm, the DC coercivity is reduced to 4.2 T, but the reversal time is doubled (Figure 2). Using frequencies close to 100 GHz, the reversal field can be substantially lowered down to 0.6 T (86% of $H_{dc}$), but the reversal time approaches 1 ns.

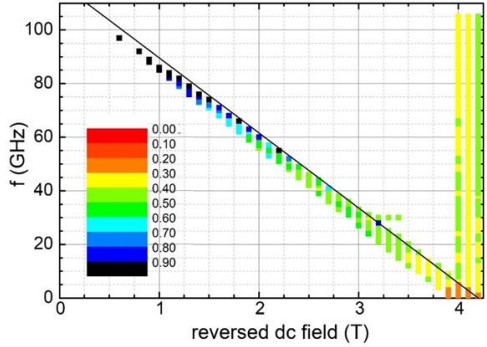

**Figure 2.** Conditions (exciting frequency vs. applied reversed field) under which reversal occurs within 1 ns for a CoPt disk with diameter 16 nm and thickness 4 nm, covered by a 2 nm $Co_3Pt$ layer. The color code represents the switching time in nanoseconds. A particular color is used for switching time, which is between the values indicated at its two edges. The solid black line represents the Kittel condition $f = \gamma\,(4.2\ T - H)$.

Next, we consider the case of having the anisotropy easy axis at θ = 45° to the applied field, which, according to the Stoner–Wohlfarth model, gives the lowest reversal field and minimal sensitivity on the switching field dependence on the easy axis distribution [38]. The data are shown in Figure 3. MAS reduction of the reversal field is achieved only at high frequencies (200 GHz) and the reduction of the switching field is only by 20%. This must be related to the different dependence of the resonance frequency on the field in the case of 45° tilting. Since, in this case, the magnetocrystalline and shape anisotropy axes do not coincide, a simple analytical expression for the FMR frequencies cannot be derived, and the resonances of the system have been found by mumax3 simulations. This was performed using the Fourier transform of the magnetization response after a sinc-function excitation. The derived frequencies agree very well with those obtained for a uniaxial Stoner–Wohlfarth particle with the DC field applied at θ = 45° to the easy axis (see Appendix A) setting $H_K$ = 12.3 T. However, the switching occurs at lower fields than the $H_K/2$ (predicted by the Stoner–Wohlfarth for θ = 45°) and the optimal frequencies are lower than the Kittel resonances of the system.

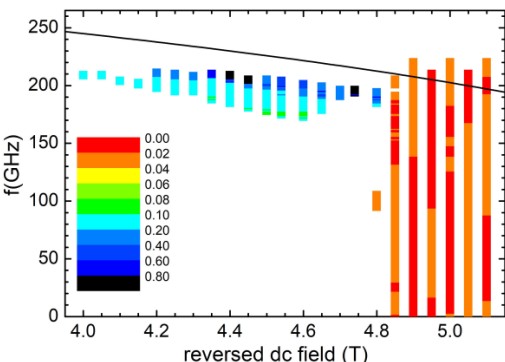

**Figure 3.** Conditions (exciting frequency vs. applied reversed field) under which reversal occurs within 1 ns for a CoPt disk with diameter 16 nm and thickness 4 nm, when the applied field is at 45° to the easy axis. The color code represents the switching time in nanoseconds. The solid black line represents the Kittel condition in this case.

In the θ = 45° case, switching at lower fields is also achieved at the expense of switching rapidity. At 5 T, the switching time is 0.02 nsec and increases to 0.15 nsec at 4 T. Combining easy axis tilting by 45°, with the addition of a softer phase (Figure 4), the switching field can be lowered to 1.8 T (by 38% of $H_{dc}$) using frequencies close to 90 GHz. At 2.95 T, the switching time is 0.05 nsec, but increases to 1 nsec at 1.8 T. One common feature of the tilted cases is that the resonance frequency does not depend sensitively on the applied DC field.

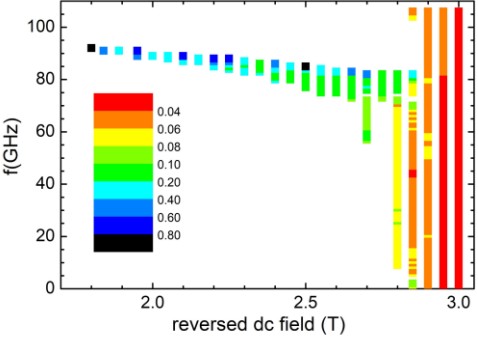

**Figure 4.** Conditions (exciting frequency vs. applied reversed field) under which reversal occurs within 1 ns for a CoPt disk with diameter 16 nm and thickness 4 nm, covered by a 2 nm Co₃Pt layer, when the applied field is at 45° to the easy axis. The color code represents the switching time in nanoseconds. A particular color is used for switching time, which is between the values indicated at its two edges.

In order to probe the effect of MAMR in a system where inhomogeneous processes govern the reversal, we consider a $D = 60$ nm CoPt disk with a thickness $t = 4$ nm (Figure 5). The frequency vs. applied field map resembles the one for the $D = 15$ nm disk; the main difference being that the reversal times are now longer. Even at 11 T, they approach 0.24 nsec. The addition of an extra 2 nm of soft phase (Figure 6) has a very drastic effect on the coercive field, but always at the expense of the switching time, which is longer than 0.59 nsec at all cases.

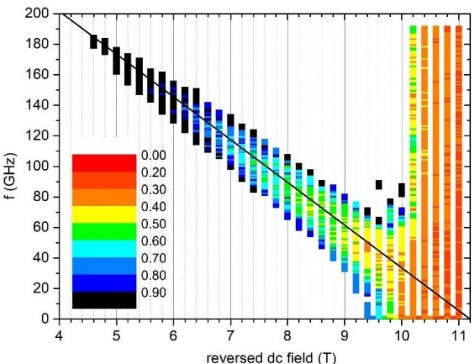

**Figure 5.** Conditions (exciting frequency vs. applied reversed field) under which reversal occurs within 1 ns for a CoPt disk with diameter $D = 60$ nm and thickness $t = 4$ nm. The color code represents the switching time in nanoseconds. A particular color is used for switching time, which is between the values indicated at its two edges. The solid black line represents the Kittel condition $f = \gamma \, (11.2 \, T - H)$.

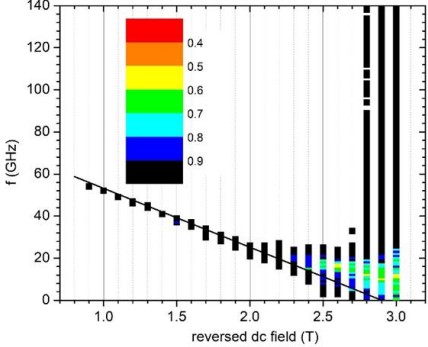

**Figure 6.** Conditions (exciting frequency vs. applied reversed field) under which reversal occurs within 1 ns for a CoPt disk with diameter $D = 60$ nm and thickness $t = 4$ nm, covered by a 2 nm $Co_3Pt$ layer. The color code represents the switching time in nanoseconds. A particular color is used for switching time, which is between the values indicated at its two edges. The solid black line represents the Kittel condition $f = \gamma \, (2.9 \, T - H)$.

## 4. Discussion and Conclusions

We have examined microwave-assisted magnetic switching (MAS) in different nanostructures as a function of applied DC field and microwave frequency. The systems chosen include (a) single-phase vs. two-phase, (b) sizes both below and above the critical size for coherent reversal, and (c) aligned and tilted anisotropy. In all the cases, the existence of microwave excitation induces the reversal at lower fields than those needed with just DC fields. The optimal frequencies are fairly close to the resonances predicted by the Kittel relations. Thus, when the field is aligned with the easy axis, the linear Kittel relation implies that the coercivity would decrease linearly with the frequency in accordance with the predictions of the rotating frame model [8]. In contrast, for anisotropy misaligned by 45°, the reduction in coercivity is limited to high frequencies. This is consistent with the form of the Kittel equation for this case.

However, this reduction in required DC field strength always comes at a cost in reversal time, as shown in Figure 7.

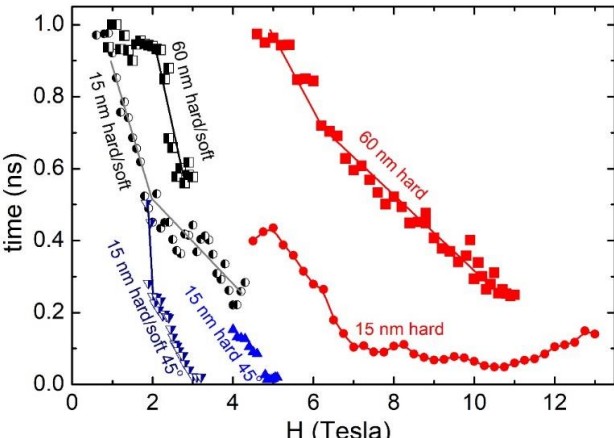

**Figure 7.** Summary of minimum magnetization reversal time using MAS, as a function of the applied reversed field, for different CoPt (hard) and CoPt/Co$_3$Pt (hard/soft) nanostructures.

For the DC reversal, in which the applied field exceeds the DC coercivity, this increase in time is easily explained within a macrospin approximation: the $H_{eff}$ that governs the dynamics is the applied field minus the effective anisotropy field. The larger their difference, the higher the frequency of the precession and the faster the dynamics. Thus, the presession frequency is initially $\gamma(H - H_K)$, but as the magnetization is reversed towards the field direction, it increases to $\gamma(H + H_K)$. Although $H_{eff}$ changes during the reversal, the sign of $H_{eff}$ and the chirality of the precession are always the same. On the other hand, using MAS, the reversal starts from fields below the anisotropy field. In this case, not only the precession frequency, but also the chirality required for the reversal, change [26] due to the sign change of the effective field $\gamma(H - H_K)$. Our simulations show that, using a fixed frequency, the resonance is limited to the initial stages of the reversal, but as the reversal proceeds, the precession frequency quickly deviates from the microwave frequency. It is reasonable then to ask if there is any point to extend the duration of the microwave pulse beyond the first stages of the reversal and specially beyond the point where the precession changes sign. However, we found no case in which the switching time was reduced by stopping the microwave pulse before the reversal is completed. The pulse rise time is also an important parameter which depends on the STO design. We examined the effect of the rise time by varying its value from 5 ps to 400 ps. The reversal time, in general, is not a monotonous function of the rise time, but presents many peaks. However, at all cases examined, the reversal time is increased with respect to the value obtained for the rise time approaching zero, but by an amount less than the rise time. The results of the simulations presented here cover diverse situations of MAS application and can provide general guidelines for the optimization of practical MAMR systems.

**Author Contributions:** Conceptualization, I.P.; methodology, C.T. and I.P.; investigation, C.T.; supervision, writing, and editing, I.P. All authors have read and agreed to the published version of the manuscript.

**Funding:** This research received no external funding.

**Institutional Review Board Statement:** Not applicable.

**Informed Consent Statement:** Not applicable.

**Data Availability Statement:** Mumax3 codes and data available on request.

**Conflicts of Interest:** The authors declare no conflict of interest.

**Appendix A**

Suppose we have a Stoner–Wohlfarth particle with a uniaxial anisotropy and easy axis in the x–z plane at θ = 45° to the z-axis and we apply a DC field $H$ in the z-direction. The free energy can be written:

$$E = -\frac{K}{2}\left(\frac{M_x}{M_s} + \frac{M_z}{M_s}\right)^2 - \mu_0 H M_z = -\frac{K}{2}(m_x + m_z)^2 - \mu_0 H M_S m_z, \quad (A1)$$

where $m_x, m_y, m_z$ are the components of the normalized magnetization. The components of the effective field are:

$$H_x^{\text{eff}} = -\frac{1}{\mu_0}\frac{\partial E}{\partial M_x} = \frac{K}{\mu_0 M_s^2}(M_x + M_z) = \frac{H_K}{2}(m_x + m_z), \quad (A2a)$$

$$H_y^{\text{eff}} = -\frac{1}{\mu_0}\frac{\partial E}{\partial M_y} = 0, \quad (A2b)$$

$$H_z^{\text{eff}} = -\frac{1}{\mu_0}\frac{\partial E}{\partial M_z} = \frac{K}{\mu_0 M_s^2}(M_x + M_z) + H = \frac{H_K}{2}(m_x + m_z) + H, \quad (A2c)$$

We suppose that, for a specific applied DC field, the magnetization precession with frequency $\omega$, denoted by the vector $\widetilde{m}(t)$, can be expanded to a first order around its static magnetization direction $m$. We, therefore, can write for each component, $\widetilde{m}_i = m_i + \delta m_i \cdot e^{i\omega t}$, $i = \text{x}, \text{y}, \text{z}$, where the $m_i$ are the static magnetization components and $\delta m_i$ are the small amplitudes of variation due to the precession.

Then, we substitute to the equation of motion $\frac{d\widetilde{m}}{dt} = -\gamma \widetilde{m} \times H_{\text{eff}}$ and keep the terms up to first order in $\delta$. The zeroth order term must vanish, since the torque vanishes when the magnetization is along its static value. This condition, if fact, gives the DC magnetization curve for a Stoner–Wohlfarth particle at θ = 45°: $Hm_x + \frac{H_K}{2}(m_x^2 - m_z^2) = 0$. The static magnetization is on the xz plane and we can use $m_x^2 + m_z^2 = 1$. The first order terms yield the condition by which the uniform mode can be calculated by the determinant:

$$\left| \begin{pmatrix} \frac{i\omega}{\gamma} & H + \frac{H_K}{2}(m_x + m_z) & 0 \\ -H - H_K m_x & \frac{i\omega}{\gamma} & +H_K m_z \\ 0 & -\frac{H_K}{2}(m_x + m_z) & \frac{i\omega}{\gamma} \end{pmatrix} \right| =, \quad (A3)$$

Finally, one gets:

$$\omega = \gamma\sqrt{H^2 + \frac{1}{2}HH_K(3m_x + m_z) + \frac{1}{2}H_K^2(m_x + m_z)^2}, \quad (A4)$$

The other two solutions being $\omega = 0$ and the negative of Equation (A4).

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
