# Peer review of "Micromagnetics of Microwave-Assisted Switching in Co-Pt-Based Nanostructures: Switching Time Minimization"

_2673-8724, doi:10.3390/magnetism3010006_

Round 1

Reviewer 1 Report

This article was submitted by Christos Thanos and Ioannis Panagitopoulos, entitled "Micromagnetics of Microwave Assisted Switching in Co-Pt based nanostructures: Switching time minimization.".

The authors investigate microwave assisted magnetic switching (MAS) for different CoPt and CoPt/Co3Pt nanostructures depending on the microwave frequency and applied DC field.

The following are my comments that I would like to share with you.

1. It would be helpful if the authors could explain why PtCo and PtCoCr structures were chosen. As a result of their high magnetocrystaline anisotropy parameters or etc.   

2. The authors use the mumax3 package to simulate micromagnetic in their study. It would be helpful to the reader if they described about the package and the program in more details in the introduction.   

3. It would be beneficial to extend the introduction section.  

4. The authors use some magnetic parameters, such as saturation magnetization and uniaxial anisotropy constants, but I have not found any reference or instructions about finding these values. 

5. Their results were explained by the Stoner-Wohlfarth model, it would be good if they provided some details about this model. 

Reviewer 2 Report

Summary

"The article describes simulations on different CoPt nanostructures to understand the effect of Microwave radiation on Magnetic Switching as a function of applied dc field and microwave frequency. The authors have carried out a comprehensive simulation study of Microwave Assisted Switching  (MAS) in CoPt & Co3Pt nanostructures of different phases, sizes, and anisotropy and its effects on the switching fields and reversal times."

Comments:

Line 46: Please mention that both reversal time and Switching time are the same before using them synonymously.

Line 70: What are the limitations in using simulating the mumaX3 package?

and please mention the error bars for the values calculated.

Figure 1: f vs reversed DC field plot. Please keep the axis names and units consistent in all the plots. Also, write how to deduce reversal time from these plots. 

Line 106: Please define all the parameters in Kittel's equation. Same with equation 1.

line 186: citation missing?
